# A Novel Space-Borne High-Resolution SAR System with the Non-Uniform Hybrid Sampling Technology for Space Targets Imaging

**Zhenghuan Xia** [1], **Shichao Jin** [1], **Fuzhan Yue** [1], **Jian Yang** [2], **Qingjun Zhang** [3], **Zhilong Zhao** [1], **Chuang Zhang** [1,*], **Wenning Gao** [1], **Tao Zhang** [1], **Yao Zhang** [1], **Xin Liu** [1] and **Tao Peng** [1]

[1] State Key Laboratory of Space-Ground Integrated Information Technology, Beijing Institute of Satellite Information Engineering, Beijing 100095, China; maxwell_xia@126.com (Z.X.); jinsc@163.com (S.J.); p15894552245@163.com (F.Y.); zhilong5552003@163.com (Z.Z.); 18612260885@163.com (W.G.); forzhangtao@163.com (T.Z.); 314603140@163.com (Y.Z.); liuxin115@foxmail.com (X.L.); pengt2021@126.com (T.P.)

[2] Department of Electronic Engineering, Tsinghua University, Beijing 100095, China; yangjian_ee@tsinghua.edu.cn

[3] China Academy of Space Technology, Beijing 100095, China; j1254621456245@163.com

\* Correspondence: zhang2012202026@126.com

**Abstract:** In order to reduce the complexity of the receiving system for wideband signals, a novel space-borne high-resolution synthetic aperture radar (SAR) system with the non-uniform hybrid sampling technology is proposed in this paper. The non-uniform hybrid sampling technology is firstly applied in the SAR imaging system for the detection of space targets. The non-uniform hybrid sampling technology is able to optimize the transmitted and receiving timing of SAR signals, reducing the requirement of the sampling rate of the analog-to-digital converter (ADC) for the reception of the wideband echoes from space targets. Meanwhile, according to the oversampling requirement of SAR imaging in the azimuth direction, a theoretical model of non-uniform hybrid sampling parameters and relative velocity between the SAR system and the space targets is established. A series of simulation experiments with different targets and different non-uniform hybrid parameters are performed, and an X-band SAR imaging experimental system is constructed to verify the effectiveness of the proposed non-uniform hybrid sampling technology. The experimental results show that the imaging resolution is better than 8 cm. When the non-uniform hybrid sampling interval is 15 us, the imaging quality is consistent with imaging results of the Nyquist real-time sampling, and it is easier to implement in the high-resolution imaging for space targets.

**Keywords:** space-borne SAR imaging; wideband SAR; non-uniform hybrid sampling; high-resolution SAR imaging; space targets

## 1. Introduction

In recent years, commercial low-orbit satellites were developed successfully. The number of low-orbit communication satellites, such as Starlink and Oneweb, will exceed ten thousand [1–3], and SAR-satellite constellations such as ICEYE and Capella SAR will exceed one thousand [4–6]. These satellites are distributed among low orbits whose height is from 300 km to 1100 km. For the safety of satellites, each satellite will be assembled with early-warning detection sensors, including microwave radars and cameras in the future. At present, space-borne SAR imaging has been successfully employed in earth observation to achieve all-time and all-weather imaging with high resolution [7]. By the reduction of antenna aperture and transmission power, the SAR also can be assembled on the satellite to make full-time imaging for surrounding space targets. Meanwhile, the mono-pulse radar has been successfully employed in searching and tracking space targets, obtaining

high-precision dynamic trajectory of space targets, and estimating the speeds and positions for SAR imaging of space targets.

Space targets are complex and diverse, such as space debris, decommissioned spacecraft, and satellites in orbit. Additionally, the orbit of low-orbit space debris is stable, the database of low-orbit space debris has been established [8,9]. Therefore, this paper focuses on high-resolution imaging of space targets, such as spacecraft and satellites. Since the size of most space targets, such as satellites and spacecraft, is between 1 m and 50 m, the resolution of radar imaging needs to be between 1 cm and 10 cm to realize the identification and classification of space targets. Therefore, the bandwidth of SAR signals should be set between 1.5 GHz and 15 GHz. Higher frequency bands, such as X-band, Ka-band, and W-band, are generally designed for the imaging system to obtain more detailed information of space targets. At present, great progress has been made in detecting ground and sea targets using SAR data, but there are still many technical difficulties in using SAR data to detect air or space targets [10]. The main detection method for air targets is inverse synthetic aperture radar (ISAR) [11], and the main detection methods for space targets are optical systems and doppler radar [12,13].

The key to high-resolution SAR imaging system is to generate and receive wideband radar signals. The frequency-doubling technology is used to generate wideband radar signals [14], but the reception of wideband radar signals still faces severe technical challenges, especially for space-borne radars. According to the Nyquist sampling theorem, the sampling rate of ADC in the SAR receiver should be more than 7 gigabit samples per second (GSPS) for the radar signal with the bandwidth of 3.5 GHz, which brings great challenges to ADC sampling devices and greatly increases the hardware cost. At present, in addition to using ultra-high-speed ADC directly, a series of methods to sample wideband signals has been proposed [15–18], which can be divided into the following three: Firstly, the de-chirp technology is able to convert wideband echoes into narrowband echoes, reducing the sampling rate of ADC [15]. However, the detection distance of de-chirp method is limited, and difficult to change. Thus, it is not suitable for high-resolution imaging of space targets at any distance; Secondly, the multi-channel receiving and stitching method in frequency domain is used to receive wideband signals, which utilizes multiple receiving channels to divide wideband echoes into multiple narrowband signals [16]. This method is complicated in hardware and stitching algorithms, so it is difficult to apply to a real-time space-borne SAR system. Thirdly, the equivalent-time sampling method employs the high-speed ADC with wideband input and programmable time delay line to repeatedly sample the wideband echo, obtaining one or more sampling points of the wideband signal in each pulse repeated period (PRP), and finally performs timing recovery in the time domain [17,18], which is able to receive wideband echo at the cost of sampling time. Although the equivalent-time sampling method is simple, the positions of the radar and target are almost stationary or quasi-static in the equivalent-time sampling period, which is difficult to meet the requirement of high-speed flight for space targets imaging.

In order to reduce the complexity of the system and achieve the real-time imaging of space targets at any distance, this paper presents a novel space-borne high-resolution SAR system with non-uniform hybrid sampling technology. The non-uniform hybrid sampling technology is firstly applied in SAR imaging system for the detection of space targets. The non-uniform hybrid sampling technology optimizes the transmitted and receiving timing of SAR signals, reducing the requirement of the sampling rate of ADC for the reception of the wideband echoes from space targets. Meanwhile, according to the oversampling requirement of SAR imaging in the azimuth direction, we establish a theoretical model of non-uniform hybrid sampling parameters and relative velocity between the SAR system and the space targets. Finally, guided by the simulated results of SAR imaging with different parameters, an X-band SAR experimental system is constructed to verify the effectiveness of the proposed non-uniform hybrid sampling technology. The experiment results show that the imaging resolution with non-uniform hybrid sampling technology is better than 8 cm, achieving high-resolution imaging for space targets at any distance.

This paper is organized as follows: Section 2 introduces the design of the space-borne X-band SAR system. Section 3 presents the proposed non-uniform hybrid sampling technology in detail. Section 4 shows the imaging results of simulation and indoor experiment. Section 5 finally gives the conclusion.

## 2. Design of the Space-Borne X-Band SAR System

The imaging scene of the proposed space-borne high-resolution SAR system for space target in this paper is shown in Figure 1. The best imaging scene of the SAR satellite platform for space target is that the low-orbit SAR satellite platform flies in the same direction as the space target, or flies in the direction of a cross-angle ($\beta_{st}$), and $|\beta_{st}|$ generally less than 30°. When the satellite and the space target fly in opposite directions, the relative velocity in the azimuth direction is as high as 15 km/s [1]. The SAR Doppler bandwidth ($B_{dop}$) is calculated by the following formula:

$$B_{dop} = 2V_{st}/D_A \tag{1}$$

where $V_{st}$ and $D_A$ are the relative velocity and the actual aperture of SAR antenna in the azimuth direction, respectively. Thus, the doppler bandwidth is about 60 KHz for a SAR antenna with 0.5 m aperture. The pulse repeated frequency (PRF) of SAR system is 84 KHz according to the requirement of 1.4 times oversampling rate. The pulse width of the SAR signal is less than 10 us, which limits the average power of SAR system, resulting in poor imaging performance of long-distance space targets. When the absolute value of $\beta_{st}$ is less than 30°, the relative velocity in the azimuth direction is generally less than 2 km/s so the doppler bandwidth is less than 8 KHz, and the PRF of SAR system is less than 12 KHz. The requirement of PRF can provide sufficient time resource for the transmission of SAR system with high average power and the non-uniform hybrid sampling of wideband echoes.

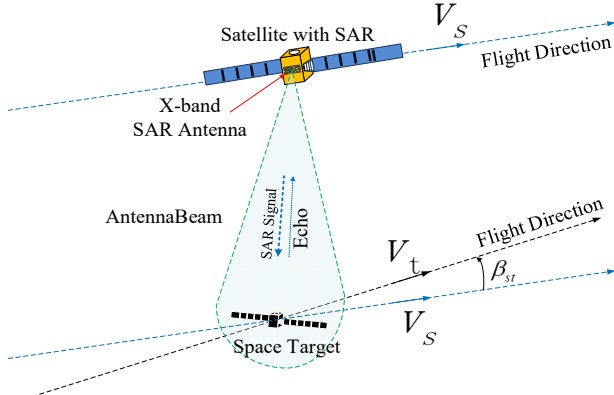

**Figure 1.** Imaging scene of space-borne SAR for space target.

The parameters of SAR systems are determined by bandwidth, PRF and imaging distance. In order to identify and classify space targets, the bandwidth of SAR signals is set to 3.5 GHz, the center frequency is set to 9.75 GHz, and the imaging resolution should reach centimeter level. According to the sampling rate requirement, the PRF should be greater than 1.4 times the doppler bandwidth. Finally, the imaging distance is from 30 km to 50 km.

The block diagram of the hardware structure is illustrated in Figure 2. The antenna of X-band SAR, signal generator of X-band SAR, circulator (transmit/receive switch), radio frequency receiver, clock generator and distributor, main control system are included. The SAR antenna is used to transmit and receive wideband SAR signals, the polarization is VV polarized, the working frequency is from 8 GHz to 11.5 GHz, and the center frequency is 9.75 GHz. The wideband chirp signals are generated by signal generator and transmitted when the rising edge of the trigger pulse arrives. The peak power of transmitted chirp

SAR signal is 0 dBm, the peak power of SAR signal is amplified to 40 dBm after the power amplifier.

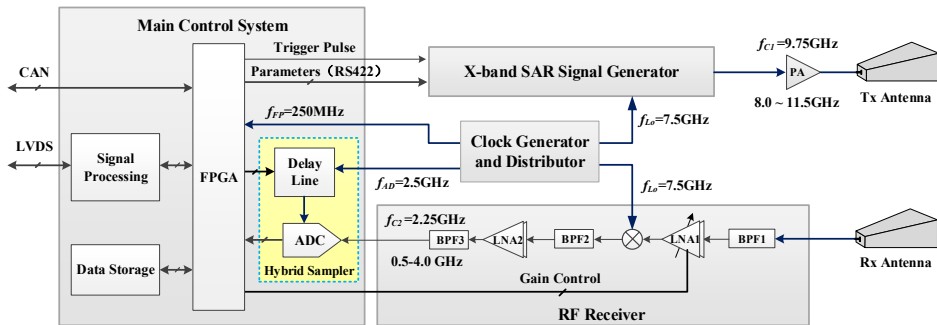

**Figure 2.** Diagram of the X-band SAR with the non-uniform hybrid.

The radio frequency receiver is designed for primary amplification, mixing, band-pass filtering, secondary amplification of the echoes. The primary amplification gain is 20 dB to 40 dB, and the gain can be controlled by the main control system. The gain of secondary amplification is 40 dB. The local frequency of the mixer is 7.5 GHz. The output frequency after mixer is from 0.5 GHz to 4.0 GHz, and the center frequency is 2.25 GHz. The entire gain of radio frequency receiver is from 50 dB to 70 dB. The clock generator and distributor provide 7.5 GHz local oscillator frequency, 2.5 GHz ADC sampling clock, and FPGA working master clock for X-band SAR signal generator, radio frequency receiver, main control system, respectively.

The main control system is used to control the parameters of the signal generator, the triggering time of launch and the gain of the radio frequency receiver. Meanwhile, it is used to control the receiving timing of the non-uniform hybrid sampler and the delay amount of the programmable delay line. It is also used to process and save the sampling echoes. Lastly, it is used to interpret and configure the work instructions and parameters of the satellite platform.

The non-uniform hybrid sampling module is mainly composed of the programmable delay line and the high-speed ADC. The minimum time-delay step of the programmable delay line is 10 ps, and the maximum time delay is 5 ns. The time delay of the programmable delay line can be periodically configured through the service provider interface (SPI) interface of the field programmable gate array (FPGA). For the high-speed ADC, the maximum input bandwidth is 6 GHz, the sampling rate is 2.6 GSPS, and the quantization resolution is 14 bit. To receive wideband echoes from 0.5 GHz to 4.0 GHz, the sampling rate of the high-speed ADC is set to 2.5 GSPS, and the time-delay step of the programmable delay line is set to 100 ps. Therefore, the equivalent sampling interval of non-uniform hybrid sampling is 100 ps, and four consecutive pulse sampling periods can complete the sampling and quantization of one wideband echo signals. The main parameters of the space-borne high-resolution X-band SAR system are listed in Table 1.

**Table 1.** The main parameters of the proposed X-band SAR system.

| Description | Value |
| --- | --- |
| Center frequency | 9.75 GHz |
| Bandwidth of the SAR signal | 3.5 GHz |
| Polarization of the antenna | VV |
| Size of the antenna | 0.28 m (A) × 0.28 m (R) |
| Peak power of the transmitted signal | 40 dBm |
| Pulse width of the transmitted signal | 100 ns~10 us |
| Gain of the RF receiver | 50~70 dB |
| Real-time sampling rate of the high-speed ADC | 2.5 GSPS |

**Table 1.** *Cont.*

| Description | Value |
|---|---|
| Quantization resolution of the ADC | 14 bits |
| Maximum input bandwidth of the high-speed ADC | 6.0 GHz |
| Equivalent-time sampling interval of the non-uniform hybrid sampling | 100 ps |
| Sensitivity of the radio frequency receiver | about −90 dBm |
| Equivalent PRF | 2 Hz~20 kHz |
| Local oscillator frequency | 7.5 GHz |
| Clock of the FPGA | 250 MHz |

## 3. The Non-Uniform Hybrid Sampling Technology

### 3.1. Timing of the Non-Uniform Hybrid Sampling Technology

As shown in Figure 3, a complete echo is obtained at $N$ PRPs with the proposed non-uniform hybrid sampling technology, and the equivalent-time pulse repeated period (E-PRP) should meet the oversampling requirement of the Doppler signal in the azimuth direction. The interval between two adjacent hybrid sampling, the number of hybrid sampling for obtaining one complete echo, and E-PRP are denoted as $t_{hs}$, $N$, and $T_{E-PRP}$, respectively. The relationship among them can be expressed as

$$T_{E-PRP} = \alpha_t \cdot N \cdot t_{hs}, \alpha_t > 1 \tag{2}$$

where $\alpha_t$ is the duty-cycle factor. Meanwhile, the time window for the reception of SAR echoes is estimated as

$$t_{hs} > t_w = (M+1)T_{ADC} \geq \frac{2W_{swath} \cdot \sin\theta_{in}}{c} + t_P \tag{3}$$

where $T_{ADC}$, $t_w$, $M$, $W_{swath}$, c, $\theta_{in}$ and $t_P$ denote the period of ADC clock, the time window of receiver, the number of ADC sampling periods with the real-time sampling rate in the time window, the imaging swath along the range direction, the speed of the light, incident angle and pulse width of the transmitted signal, respectively. Thus, the total number of the sampling points for each echo with the non-uniform hybrid sampling technology is $N_S = M \cdot N$, and the equivalent-time sampling interval is $t_{int} = T_{ADC}/N$. For the spaceborne SAR system, the ADC real-time sampling rate is 2.5 GSPS, the sampling period is 400 ps, and the time-delay step of 100 ps can be achieved through the programmable delay line. In other words, the equivalent-time sampling interval is 100 ps, and the equivalent-time sampling rate is 10 GSPS, which can meet the oversampling requirements of the echo with a bandwidth of 3.5 GHz. A complete wideband echo can be obtained after four consecutive periods of hybrid sampling.

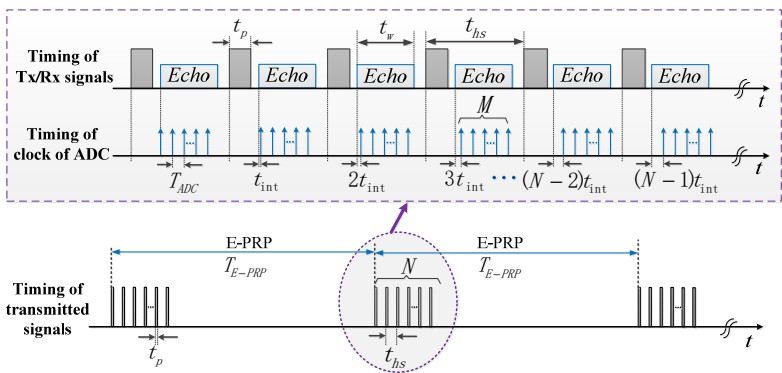

**Figure 3.** Timing of the non-uniform hybrid sampling technology.

### 3.2. Analysis of the Non-Uniform Hybrid Sampling Technology for High-Resolution SAR Imaging

Due to the high relative velocity between the space-borne X-band SAR and space targets, the phase and time of the SAR echoes will be disturbed when the proposed non-uniform hybrid sampling technology is applied to receive wideband SAR echoes. The beam angle in the azimuth direction is narrow, so the maximum time disturbance appears at the edge of the antenna beam. As shown in Figure 4, due to the relative motion between the space-borne X-band SAR and space targets, the maximum wave-path difference can be expressed as [19]

$$\Delta R_i \approx v_{st} \cdot N \cdot t_{hs} \cdot \sin(\theta_A/2) \approx \frac{v_{st} \cdot N \cdot t_{hs} \cdot \lambda_c}{2 D_A} \tag{4}$$

where $v_{st}$, $\lambda_c$, and $\theta_A$ are the relative velocity between the space-borne X-band SAR and space targets, the wavelength of the center frequency, and the beam angle, respectively. In order to reduce the side lobes of the SAR echoes in the range direction, the maximum time disturbance should be smaller than the equivalent-time sampling interval of the non-uniform hybrid sampling, which is estimated as

$$\frac{v_{st} \cdot N \cdot t_{hs} \cdot \lambda_c}{D_A \cdot c} \leq \frac{t_{int}}{P} \leq \frac{1}{P \cdot \eta_r \cdot B_w}, \ P \geq 2, \eta_r \geq 2 \tag{5}$$

where $B_w$, $\eta_r$ and $P$ are the bandwidth of SAR signal, the oversampling rate in the range direction, and the limited factor for the time disturbance, respectively. Then, Equation (5) can be organized as

$$\frac{D_A \cdot f_c}{v_{st} \cdot N \cdot t_{hs} \cdot \eta_r \cdot B_w} \geq P \tag{6}$$

where $f_c$ is the center frequency of the X-band SAR signals. Obviously, Equation (6) can be satisfied easily. If the relative velocity is not too large, such as 1 km/s, Equation (6) is easy to meet in engineering applications.

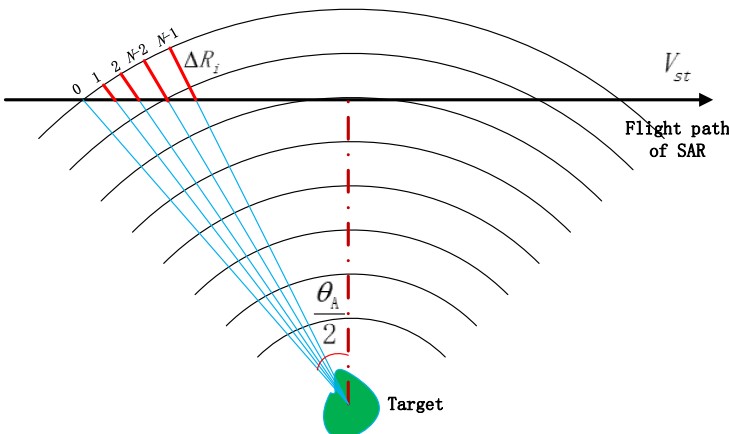

**Figure 4.** Analysis of the non-uniform hybrid sampling technology for the mini-SAR.

On the other hand, to reduce the side-lobes of the SAR echoes in azimuth direction, the maximum phase perturbation needs to be limited as

$$\frac{2\pi \cdot \Delta R_i}{\lambda_c} = \frac{\pi \cdot v_{st} \cdot N \cdot t_{hs}}{D_A} \leq \frac{\pi}{Q}, Q \geq 2 \tag{7}$$

where $Q$ is the qualification factor for the phase disturbance of SAR echoes in a complete synthetic aperture time. Equation (7) is further simplified to

$$\frac{D_A}{v_{st} \cdot N \cdot t_{hs}} \geq Q \tag{8}$$

Likewise, Equation (8) is easy to meet when the relative velocity is not too high. It is worth emphasizing that the sampling requirement of Equation (8) is more difficult to meet than that of the Equation (6) when the space-borne SAR is a typical narrowband system. However, the space-borne X-band SAR proposed in this paper is a typical wideband system, which can easily meet the sampling requirement of Equation (8). According to Equations (3) and (8), the interval between two adjacent hybrid sampling ($t_{hs}$) should meet the following condition:

$$\frac{2W_{swath} \cdot \sin \theta_{in}}{c} + t_P < t_{hs} \leq \frac{D_A}{v_{st} \cdot N \cdot Q} \tag{9}$$

It can be concluded that the interval ($t_{hs}$) should be small enough to reduce the time and phase disturbance. Meanwhile, the interval ($t_{hs}$) should also be large enough to receive the backscattered signals of all the targets located in the entire imaging swath.

## 4. Results and Discussion

### 4.1. Simulated Results of High-Resolution SAR Imaging

The simulation is carried out in order to analyze the degree of deterioration of SAR image caused by the relative motion between the space-borne X-band SAR and space targets. The simulated parameters of non-uniform hybrid sampling technology are given in Table 2. The range of hybrid samplings interval ($t_{hs}$) is calculated based on Equation (9), and the parameters setting are shown in Table 3, indicating that the hybrid sampling interval ($t_{hs}$) decreases as the qualification factor ($Q$) increases. According to the parameters of the SAR system, the hybrid sampling intervals ($t_{hs}$) for different qualification factors ($Q$) are set as 50 us, 28 us and 15 us. Simulated results of space-borne high-resolution SAR imaging for single point target are shown in Figure 5. The simulation results show that all the targets are well focused in the range direction, while the focusing performance in the azimuth direction significantly deteriorates with the increase in the hybrid sampling interval ($t_{hs}$). Considering the system characteristics of ultra-wideband signal and long synthetic aperture time, the back projection (BP) algorithm is used to image the target [20]. The simulation raw SAR data with 2.5 GSPS sampling rate are firstly processed by hybrid sampling technology to form the pre-processed data with 10 GSPS sampling rate, and then focused into a two-dimensional SAR image after pulse compression and back projection processing. Figure 5a is the imaging result of single point target with Nyquist real-time sampling (the real-time ADC sampling rate is 10 GSPS), and there is no phase disturbance in slow time. Figure 5b–d shows the imaging results with hybrid sampling intervals of 15 us, 28 us and 50 us, respectively. The quantitative analysis of imaging quality for single-point target is compared in Table 4, including impulse response width (IRW), peak side lobe ratio (PSLR) and integrated side lobe ratio (ISLR) in both the range (R) and azimuth (A) directions [21]. The imaging quality with a hybrid sampling interval of 15 us is comparable to the imaging results using Nyquist real-time sampling, whereas the azimuth imaging quality with a hybrid sampling interval of 28 us and 50 us deteriorates in the azimuth direction, especially for the 50 us interval.

**Table 2.** Simulated parameters of the proposed non-uniform hybrid sampling technology.

| Description | Value |
| --- | --- |
| Relative speed | 1 km/s |
| Size of the antenna | 0.14 m (A) × 0.14 m (R) |
| Equivalent PRF | 20 kHz |
| Pulse width of the transmitted signal | $t_P$ = 2 us |
| Center frequency | 9.75 GHz |
| Bandwidth of the SAR signal | 3.5 GHz |

**Table 2.** *Cont.*

| Description | Value |
|---|---|
| Real-time sampling rate of the high-speed ADC | 2.5 GSPS |
| The number of hybrid sampling for obtaining one entire echo | $N = 4$ |
| Equivalent-time sampling interval | $t_{int} = 100$ ps |
| Qualification factor for the phase disturbance | $Q = 2/3, 6/5, 2$ |

**Table 3.** The interval between two adjacent hybrid samplings.

| Qualification Factor | Range of Hybrid Sampling Interval (us) | Hybrid Sampling Interval (us) |
|---|---|---|
| $Q = 2/3$ | $2.20 < t_{hs} < 52.50$ | $t_{hs} = 50$ |
| $Q = 6/5$ | $2.20 < t_{hs} < 29.17$ | $t_{hs} = 28$ |
| $Q = 2$ | $2.20 < t_{hs} < 17.50$ | $t_{hs} = 15$ |

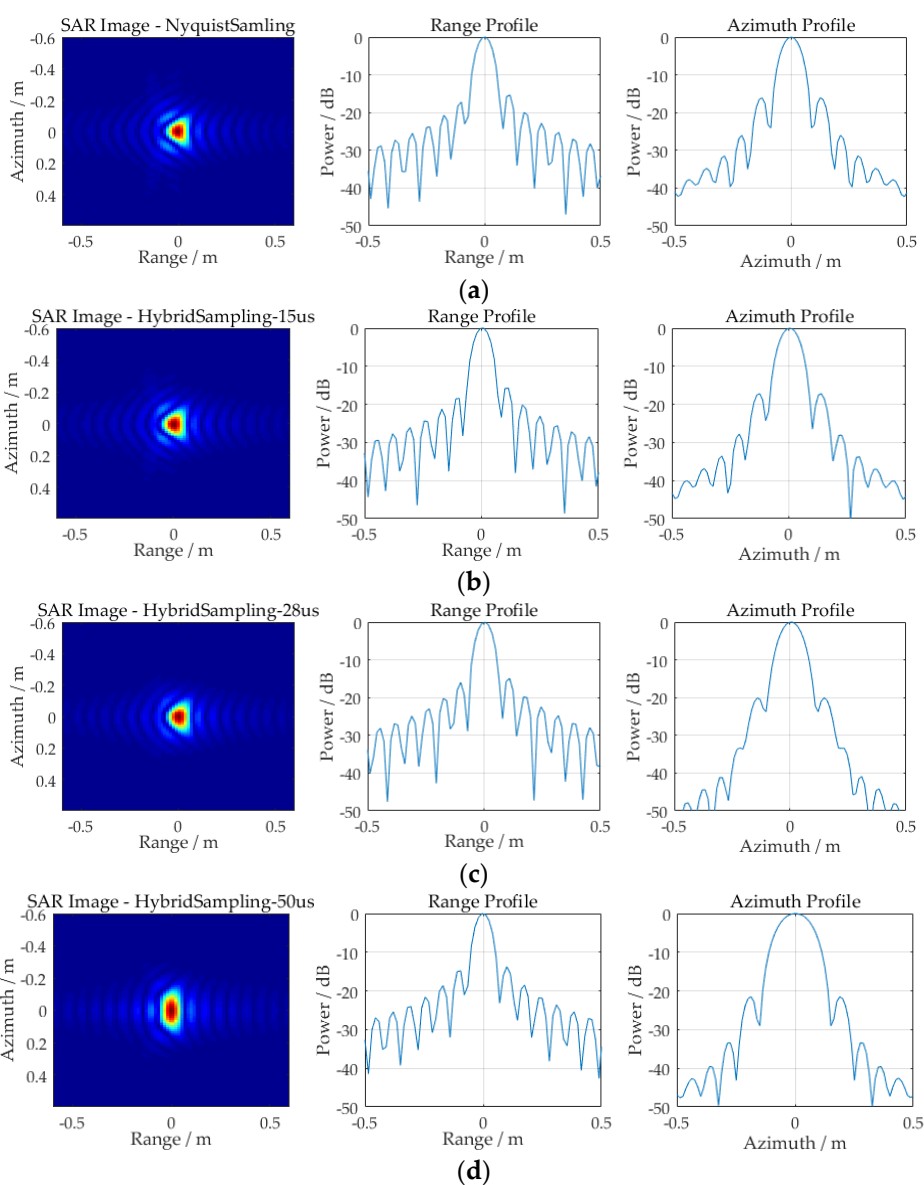

**Figure 5.** The SAR simulated results of single point target with different $Q$ factors: (**a**) Nyquist real-time sampling; (**b**) $Q = 2$, $t_{hs} = 15$ us; (**c**) $Q = 6/5$, $t_{hs} = 28$ us; (**d**) $Q = 2/3$, $t_{hs} = 50$ us.

**Table 4.** The analysis of imaging quality for single point target with different factors.

| Quality Parameters | $Q = 2/3$ ($t_{hs} = 50$ us) | $Q = 6/5$ ($t_{hs} = 28$ us) | $Q = 2$ ($t_{hs} = 15$ us) | Nyquist Sampling |
|---|---|---|---|---|
| IRW (R) | 7.00 cm | 7.10 cm | 7.10 cm | 7.50 cm |
| PSLR (R) | −9.1137 dB | −14.8800 dB | −15.7200 dB | −15.4531 dB |
| ISLR (R) | −13.7754 dB | −15.2739 dB | −15.8848 dB | −16.2579 dB |
| IRW (A) | 15.00 cm | 9.00 cm | 7.80 cm | 7.50 cm |
| PSLR (A) | −25.3909 dB | −20.0400 dB | −17.2000 dB | −13.9207 dB |
| ISLR (A) | −30.0375 dB | −19.9563 dB | −18.8613 dB | −18.0941 dB |

In addition, simulation is designed for point array targets with different qualification factors. Figure 6 shows the point array targets with an interval of 15 cm. The imaging results are demonstrated in Figure 7. Figure 7a is the imaging results using Nyquist real-time sampling, and Figure 7b–d are the imaging results using the proposed non-uniform hybrid sampling technology with 15 us, 28 us and 50 us intervals, respectively. Comparing Figure 7a,b, point targets can clearly be distinguished in both the range and azimuth direction. However, the imaging result in Figure 7c,d reveals that all points in the range direction could be clearly distinguished, but the points in the azimuth direction cannot be clearly distinguished. Therefore, when the non-uniform sampling interval is set to 15 us, the imaging quality is consistent with the imaging result with Nyquist real-time sampling. When the non-uniform sampling interval is set to 28 us and 50 us, the azimuth imaging quality is obviously degraded.

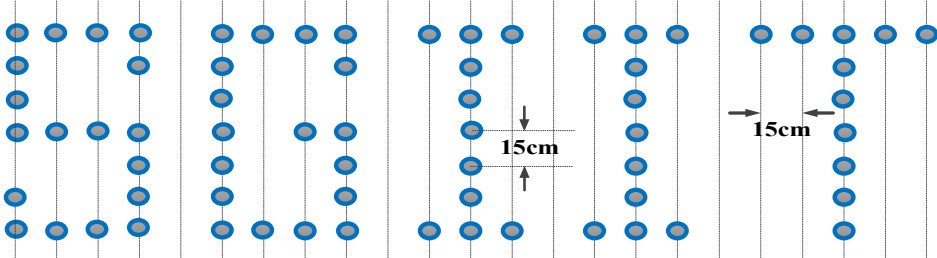

**Figure 6.** Imaging scene of point targets with the interval of 15 cm.

### 4.2. Indoor Experimental Results of High-Resolution SAR Imaging

A series of indoor SAR imaging experiments with different qualification factors are performed to further verify the effectiveness of the proposed novel space-borne high-resolution SAR system, using non-uniform hybrid sampling technology. The experimental scene is shown in Figure 8, and the satellite model is used as the imaging target. The azimuth size of the satellite is 1.5 m (including the satellite body and two solar wings), the distance between the front surface of the satellite body and that of the solar wings is about 20 cm. The transmitted and receiving antennas of the imaging system are separated and move at a constant speed along the orbit, which is laid along the azimuth direction. The experimental parameters are listed in Tables 5 and 6.

The SAR imaging result using Nyquist real-time sampling by oscilloscopes (Tektrnoix, DPO71254C) is shown in Figure 9a. The satellite body and the front surface of the two solar wings can be clearly distinguished from the enlarged figure. The imaging results using the proposed non-uniform hybrid sampling technology with qualification factors 2, 6/5 and 2/3 are displayed in Figure 9b–d, respectively. The actual sampling rate is 2.5 GSPS, so the corresponding upper limit of non-uniform hybrid sampling pulse interval is 0.66 s, 1.09 s and 1.97 s, respectively. The imaging result in Figure 9b shows that the three front surfaces are clearly distinguished, and the imaging quality is consistent with that in Figure 9a. Similar to the simulation results, the imaging results in Figure 9c,d are obviously defocused in the azimuth direction, the front surface of the satellite body cannot be clearly distinguished, and the front surfaces of the two solar wings also become blurred.

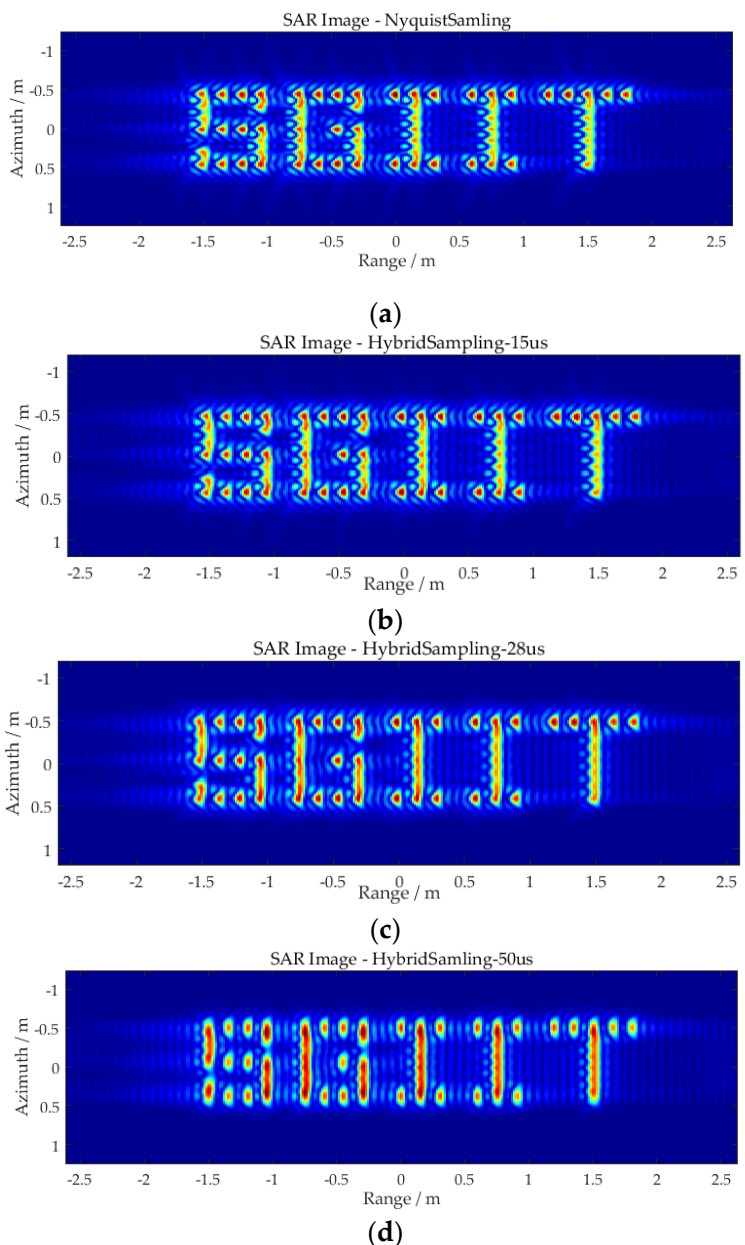

**Figure 7.** The SAR simulated results of point array targets with different Q factors: (**a**) Nyquist real-time sampling; (**b**) $Q = 2$, $t_{hs} = 15$ us; (**c**) $Q = 6/5$, $t_{hs} = 28$ us; (**d**) $Q = 2/3$, $t_{hs} = 50$ us.

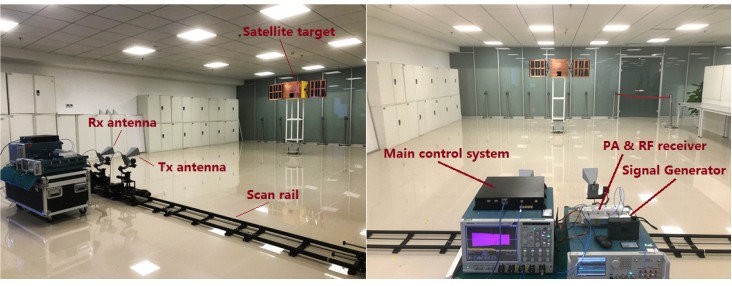

**Figure 8.** Imaging scene of the satellite target.

**Table 5.** Experimental parameters of the proposed non-uniform hybrid sampling technology.

| Description | Value |
|---|---|
| Slant range | 5.532 m |
| Pulse width of the transmitted signal | 0.12 us |
| the size of antenna in the azimuth direction | 14 cm |
| Velocity of Tx/Rx antenna | 0.0266 m/s |
| Pulse repetition frequency | 4.57 Hz |
| Qualification factor for the phase disturbance | 2, 6/5, 2/3 |
| Hybrid sampling interval | $\leq$0.66 s, $\leq$1.10 s, $\leq$1.98 s |

**Table 6.** The interval between two adjacent hybrid samplings in experiment.

| Qualification Factor | Range of Hybrid Sampling Interval (us) | Hybrid Sampling Interval (us) |
|---|---|---|
| $Q = 2/3$ | $2.20 < t_{hs} < 1.98$ | $t_{hs} = 1.97$ |
| $Q = 6/5$ | $2.20 < t_{hs} < 1.10$ | $t_{hs} = 1.09$ |
| $Q = 2$ | $2.20 < t_{hs} < 0.66$ | $t_{hs} = 0.66$ |

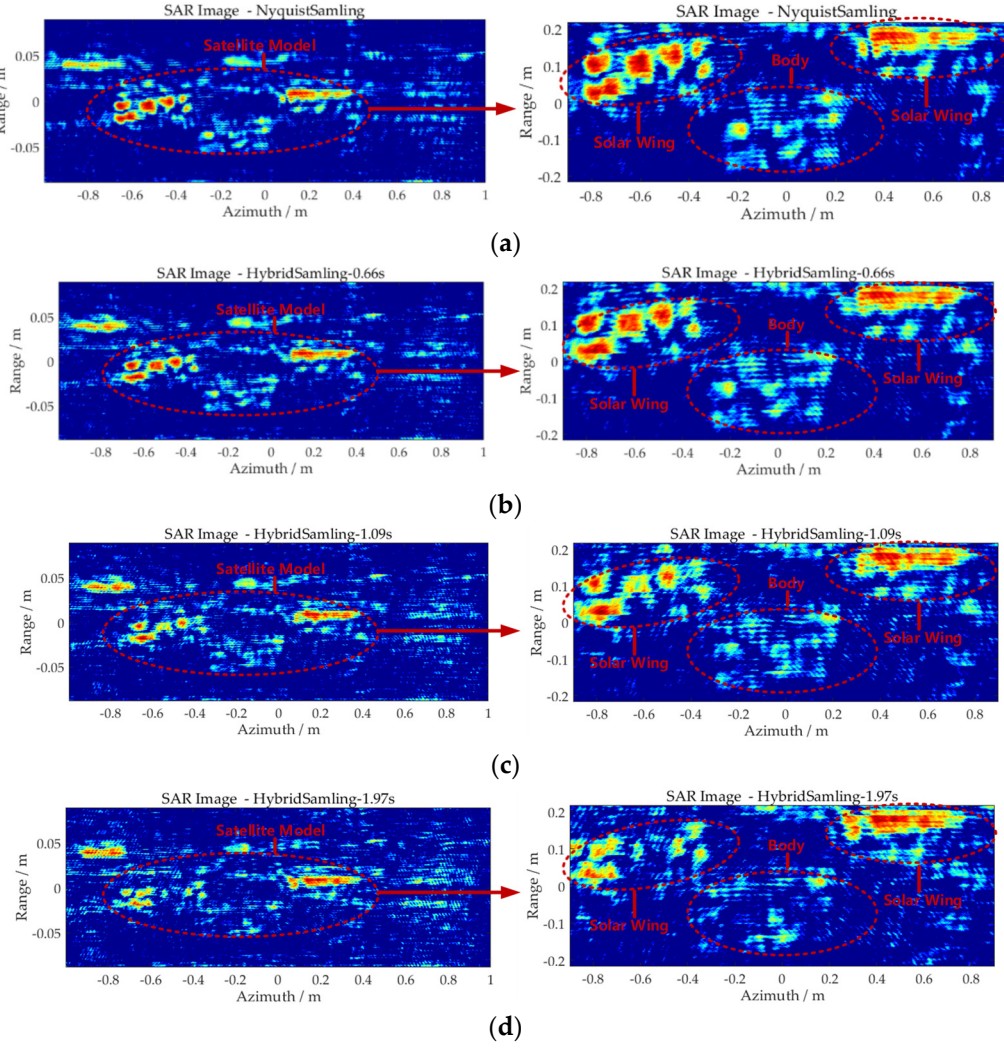

**Figure 9.** The SAR experimental imaging results of satellite model target with different Q factors: (**a**) Nyquist real-time sampling; (**b**) $Q = 2$, $t_{hs} = 0.66$ s; (**c**) $Q = 6/5$, $t_{hs} = 1.09$ s; (**d**) $Q = 2/3$, $t_{hs} = 1.97$ s.

Moreover, to quantitatively analyze the imaging performance, the sharpness metric parameter ($S$) is quoted to predict the relative amount of blurriness in images [22], which is defined as the ratio:

$$S = \frac{\sum_{(x,y)} |\sigma(x,y)|^4}{\left|\sum_{(x,y)} |\sigma(x,y)|^2\right|^2} \tag{10}$$

where $\sigma(x, y)$ is the amplitude of the complex image at point $(x, y)$. When the image becomes blurry, the value of the sharpness metric decreases. According to Equation (10), the sharpness metric for Figure 9a–d is calculated to be $1.436 \times 10^{-6}$, $1.433 \times 10^{-6}$, $1.416 \times 10^{-6}$ and $1.387 \times 10^{-6}$, respectively. Compared to Figure 9a, the value of the sharpness metric for Figure 9c to Figure 9d is decreased 0.21%, 1.39% and 3.41%, respectively. The calculated results are consistent with the visual perception of images, and it shows that the imaging performance decreases with the decrease in the value of the sharpness metric.

## 5. Conclusions

In this paper, a novel space-borne high-resolution SAR system with non-uniform hybrid sampling technology is proposed. The non-uniform hybrid sampling technology is firstly applied in SAR imaging system for the detection of space targets. By means of optimizing the transmitted and receiving timing of SAR signals, the non-uniform hybrid sampling technology is able to reduce the requirement of the sampling rate of ADC for the reception of the wideband echoes from space targets. In the paper, compared with the requirement of Nyquist sampling theorem, the sampling rate of ADC can be reduced four times through the non-uniform hybrid sampling technology. Simulated results with different targets and different non-uniform hybrid sampling parameters were compared, indicating that the imaging quality using non-uniform hybrid sampling with an interval of 15 us is consistent with imaging result of the Nyquist real-time sampling. An X-band SAR imaging experimental system was constructed, and the experimental results showed that the imaging resolution of the SAR system is better than 8 cm, satisfying the requirement of high-resolution imaging for space targets.

**Author Contributions:** Conceptualization, Z.X. and S.J.; Data curation, Z.Z. and C.Z.; Formal analysis, C.Z.; Funding acquisition, Z.X.; Investigation, Z.X. and C.Z.; Methodology, Z.X. and F.Y.; Project administration, Z.X.; Resources, W.G. and T.Z.; Software, J.Y. and Z.Z.; Supervision, Y.Z.; Validation, Z.X. and Q.Z.; Visualization, X.L. and T.P.; Writing—original draft, C.Z.; Writing—review & editing, Z.X., J.Y., C.Z. and X.L. All authors have read and agreed to the published version of the manuscript.

**Funding:** This research work was supported by Innovation Funds of Equipment Pre-Research under Grant JZX7Y20190253036001 and JZX7Y20190253040601.

**Conflicts of Interest:** The authors declare no conflict of interest.

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
