# Peer review of "A Novel Space-Borne High-Resolution SAR System with the Non-Uniform Hybrid Sampling Technology for Space Targets Imaging"

_applsci, doi:10.3390/app12104848_

Round 1

Reviewer 1 Report

The paper proposes a novel method that performs non-uniform sampling of the radar data. The paper's introduction misses a deeper analysis of the literature and a clearer presentation of the proposed method's novelties and benefits. Sections 2 and 3 describe the proposed method and require a better explanation in some parts to properly describe the method. Section 4 requires more information, including a description of both datasets (description of the simulation/acquisition, number of radar data, preprocessing, etc.) and the experiment design to validate the proposed method. Regarding the first dataset, I would also expect more simulations with different sampling times to better understand the limitation of the proposed technique. Regarding the second dataset, there is no quantitative analysis, and, also for this dataset, I would expect more experiments to validate the proposed method.

Generally, the English language is poor. It is generally hard to read and has extremely poor technical terminology. In addition, the paragraphs miss cohesion—the sentences are not connected, and some of them seem in the wrong section/paragraph.

Abstract and Introduction

The introduction and the abstracts are confusing. Connections between sentences in the same paragraph are missing.

The paper misses the literature analysis on space debris detection and, in general, target detection with SAR data.

The analysis on the 'receiving methods' misses references and should be better introduced.

The novelty of the paper is not clearly introduced in the introduction. The paper also misses a paragraph

Section 2 and 3

Imaging scene or acquisition geometry?

It is not clear whether sentences in lines 93-95' are an assumption or not.
'The best imaging scene of the SAR satellite 93 platform for space target is that the SAR satellite platform and space target both fly in the 94 same directions, or fly in the directions with a small cross-angle (?st), which generally 95 means |?st|<30°.

This sentence requires a justification or reference: 'When the absolute value of ?st is less than 30°, the relative speed in the azimuth direction is generally less than 2 km/s.'

'Composition of the space-borne high-resolution SAR system' should be rewritten in something like 'The block scheme of the hardware structure'.

The paper misses an explanation on the choice of the parameters in table 1.

Add some details on the system receiver. Also, a band of 3.5Ghz is extremely large and hard to handle. How is this done?

It is not clear if there is an assumption on the absolute and relative speeds of the SAR and the targets.

Section 4

The sentence in line 211 is not finished.

From the Table 3, it could be seen that the hybrid sampling interval (?hs) will decrease as the qualification factor (?) increases. -> from the table or the equation?

Missing connection with the surrounding paragrpf for sentence: 'The traditional Back Projection (BP) algorithm is used in 216 imaging processing [17].'

The paper misses a description of the simulations performed for the first set of experiments.

The post-processing of the data (for both datasets) is not explained.

Table4: The quality parameters of table 4 are not defined mathematically. Moreover, it is unclear how many/which data the values in tab 4 are computed. I would also expect more hybrid sampling times to better understand the proposed method's limits.

The paper misses the comparison with state-of-the-art methods.

Better structure the first dataset experiments.

The session misses a qualitative analysis of the results on the second dataset. It would also be interesting to see results with other hybrid sampling times and Tx/Rx antenna velocity.

Section 5

The concluding section should be improved by citing both datasets and adding the benefits of the proposed method.

Reviewer 2 Report

  This article proposes a novel space-borne high-resolution SAR to reduce the complexity of receiving system by means of non-uniform hybrid sampling technology.

The paper is well written. The proposed method was verified by simulation and experimentally and satisfies the requirement for high-resolution imaging for space target.   

However, the authors have not positioned their work in relation to the state of the art. The authors should give their contribution compared to the related work.

Reviewer 3 Report

According to the article [Manuscript ID: applsci-1643564 (A novel space-borne high-resolution SAR system with the non-uniform hybrid sampling technology for the imaging of space targets)], this study aims to reduce the complexity of receiving system for the wideband signals of the SAR system based on a novel space-borne high-resolution synthetic aperture radar with the non-uniform hybrid sampling technology for the imaging of space targets. Based on this study, the proposed sensor gives an imaging resolution of better than 8 cm, which can satisfy the requirement of high-resolution imaging for space targets.

The general review:

The abstract is shorter than expected, and the authors are obliged to add the added value of their proposed sensors. What is the really different between the suggested and the known sensor?

Also, all the sections must have the same, why the users will prefer the proposed sensor (e.g. the expected price, the covered area, time of getting the images, the required targets) more than the used one.

The final decision

The authors are obliged to add more about the new sensor compared with the known sensor especially the coordinates system, the technical work method of the proposed sensor, so I suggest major revision for the article before the re-submitting process

Round 2

Reviewer 1 Report

Second review for ‘A novel space-borne high-resolution SAR system with the non-uniform hybrid sampling technology for the imaging of space targets’.

The paper is considerably improved compared to the previous version—the introduction has a more comprehensive analysis of the literature. In addition, the proposed method’s description is clearer, and the experimental results look complete. Please find minor comments below:

  • Consider shortening the title.
  • Line 46, the paper contributions are introduced before the state of the art analysis. This is quite strange, given that the literature analysis describes the gap that a paper should fill.
  • Line76 and 171, there is an error reference not found.
  • The titles in fig 5 are hard to read. Are they needed? Also, the label for the x-axis is cut.
  • Figs on pages 7 to 11/12 in the track change manuscript have the same labels (Fig 5).

Reviewer 2 Report

the authors addressed my remarks 
I suggest accept the paper for publication

Author Response

Dear reviewer

Thanks very much for taking your time to review this manuscript. I really appreciate all your comments and suggestions! Thanks again!
Wish you the best of luck!

Sincerely,
Chuang Zhang

Reviewer 3 Report

The article is finely revisioned, so I recommend accept in its present form.

Author Response

(The authors gave the same response as above.)
